

# Presence of microplastics and microparticles in Oregon Black Rockfish sampled near marine reserve areas

Katherine S. Lasdin[1,2], Madison Arnold[3], Anika Agrawal[4],
H. William Fennie[5,6,7], Kirsten Grorud-Colvert[5], Su Sponaugle[5,7],
Lindsay Aylesworth[8], Scott Heppell[2] and Susanne M. Brander[9]

[1] School of Aquatic and Fishery Sciences, University of Washington, Seattle, WA, United States
[2] Department of Fisheries, Wildlife, and Conservation Sciences, Oregon State University, Corvallis, Oregon, United States
[3] Department of Environmental Molecular Toxicology, Oregon State University, Corvallis, OR, United States
[4] Natural Resources and the Environment, University of Connecticut, Storrs, CT, United States
[5] Department of Integrative Biology, Oregon State University, Corvallis, OR, United States
[6] Fisheries Resources Division, Southwest Fisheries Science Center, National Marine Fisheries Service, National Oceanic and Atmospheric and Administration, La Jolla, CA, USA
[7] Hatfield Marine Science Center, Newport, OR, USA
[8] Oregon Department of Fish and Wildlife, Newport, OR, USA
[9] Coastal Oregon Marine Experiment Station, Oregon State University, Newport, Oregon, United States

Corresponding author
Katherine S. Lasdin, klasdin@uw.edu

## ABSTRACT

Measuring the spatial distribution of microparticles which include synthetic, semi-synthetic, and anthropogenic particles is critical to understanding their potential negative impacts on species. This is particularly important in the context of microplastics, which are a form of microparticle that are prevalent in the marine environment. To facilitate a better understanding of microparticle occurrence, including microplastics, we sampled subadult and young juvenile Black Rockfish (*Sebastes melanops*) at multiple Oregon coast sites, and their gastrointestinal tracts were analyzed to identify ingested microparticles. Of the subadult rockfish, one or more microparticles were found in the GI tract of 93.1% of the fish and were present in fish from Newport, and near four of five marine reserves. In the juveniles, 92% of the fish had ingested one or more microparticles from the area of Cape Foulweather, a comparison area, and Otter Rock, a marine reserve. The subadults had an average of 7.31 (average background = 5) microparticles detected, while the juveniles had 4.21 (average background = 1.8). In both the subadult and juvenile fish, approximately 12% of the microparticles were identified as synthetic using micro-Fourier Infrared Spectroscopy (micro-FTIR). Fibers were the most prevalent morphology identified, and verified microparticle contamination was a complex mixture of synthetic (~12% for subadults and juveniles), anthropogenic (~87% for subadults and 85.5% for juveniles), and natural (*e.g.*, fur) materials (~0.7% for subadults and ~2.4% for juveniles). Similarities in exposure types (particle morphology, particle number) across life stages, coupled with statistical differences in exposure levels at several locations for subadult fish, suggest the potential influence of nearshore oceanographic patterns on microparticle distribution. A deeper understanding of the

impact microplastics have on an important fishery such as those for *S. melanops*, will contribute to our ability to accurately assess risk to both wildlife and humans.

# INTRODUCTION

Disposal of consumer waste and its degradation has led to an abundance of microscopic particles, including plastic, dispersing throughout air, water, sediments, terrestrial habitats, and food sources (*e.g.*, (*Bouwmeester, Hollman & Peters, 2015*; *Rochman et al., 2015*; *Andrady, 2017*; *Kanhai et al., 2018*; *Jamieson et al., 2019*; *Allen et al., 2019*; *Cox et al., 2019*). Microparticles, which include microplastics (*Miller et al., 2021*) (<5 mm) are a classification of contaminants (*Rochman et al., 2019*; *Miller et al., 2021*) that encompass a variety of materials and morphologies. Microplastics have been found across terrestrial and aquatic environments, as deep and secluded as the Mariana Trench (*Chiba et al., 2018*). They are present in sediments (*Peng et al., 2020*; *Xue et al., 2020*), biota (*Rochman et al., 2015*; *Cox et al., 2019*; *Baechler et al., 2020b*; *Barboza et al., 2020*), and the atmosphere (*Brahney et al., 2021*).

Microplastics are internalized by many organisms including invertebrates (*Axworthy & Padilla-Gamiño, 2019*; *Baechler et al., 2020a*; *Horn, Granek & Steele, 2020*), fish (*Rochman et al., 2015*; *Steer et al., 2017*; *Caruso et al., 2018*; *Yin et al., 2018*; *Brandts et al., 2018*; *Yin et al., 2019*; *Nanninga, Scott & Manica, 2020*; *Zitouni et al., 2021*), marine mammals (*Nelms et al., 2018*) and seabirds (*Caldwell et al., 2022*). When these plastics are internalized by fishes, they can impact multiple body functions including swimming (*Pannetier et al., 2020*; *Siddiqui et al., 2022*), stress (*Jacob et al., 2020*), energy budgets (*Cole et al., 2015*), gene expression (*Wang et al., 2019*; *Li et al., 2020b*), and to a degree, survival (*Pannetier et al., 2020*; *Assas et al., 2020*), among others. Laboratory studies in juvenile fish exposed to microplastics have demonstrated decreased growth and survival, reduced body mass and length, as well as physiological stress, which have implications for the health of populations (*Critchell & Hoogenboom, 2018*; *Naidoo & Glassom, 2019*; *Athey et al., 2020b*; *Stienbarger et al., 2021*; *Siddiqui et al., 2022*).

Fish satisfy approximately 17% of the global animal-based protein demand (*FAO, 2016*). Due to the importance of fished species and their contribution to the human diet, research into the effects of microplastics on harvested species is necessary (*Baechler et al., 2020b*). One such important harvested species in Oregon is Black Rockfish (*Sebastes melanops*). Accounting for the highest recreational quota of groundfish caught in 2021, these fish are an important recreational resource (*Oregon Department of Fish & Wildlife, Marine Resources Program, 2021*).

Adult Black Rockfish are opportunistic benthic and water column feeders (*Love, Yoklavich & Thorsteinson, 2002*; *Doran, 2020*), whereas juveniles have both a benthic and pelagic stage (*Love, Yoklavich & Thorsteinson, 2002*).

Due to their high consumption by humans, ecological importance and cultural importance along the Pacific U.S. coast, Black Rockfish are an ideal species to investigate life stage-specific microplastic occurrence in and to identify whether microplastic abundance, morphology or color differs ontogenetically (life stage) and spatially (geographic location). Further, to the best of our knowledge, this is the first microplastic ingestion study performed in nearshore fishes along the coast of Oregon, and the first study conducted in the vicinity of multiple marine reserves in North America.

To effectively manage and conserve species in marine environments, it is necessary to fully understand the occurrence of microplastic and microparticle pollution in biota; yet more data are needed regarding microplastic internalization in marine fishes, particularly those consumed as seafood (*Baechler et al., 2020b*). Filling in these data gaps is important for understanding relevant environmental concentrations of plastics and developing management strategies for fisheries. Overall, the aims of this study were to determine the distribution of microparticles, including microplastics ingestion *via* occurrence in Black Rockfish and to identify whether microplastic abundance, morphology, or color differs ontogenetically (life stage) and spatially (geographic location). Therefore, we hypothesize that there will be both macro- and microparticles in the subadult fish and microparticles in the juvenile fish, that there may be differences in the morphology of microparticles ingested by fish at different life stages, and that there will be microplastics in both life stages. Furthermore, we predict that there will be minimal spatial variation in particle ingestion by site, due to oceanographical processes.

## MATERIALS AND METHODS

### Subadult rockfish collection

The subadult rockfish were sampled in nearshore waters off the Oregon Coast from two distinct sampling efforts: samples collected in cooperation with a local recreational charter fishing company in the port town of Newport, Oregon (Charter Fish) and samples collected by the Oregon Department of Fish and Wildlife (ODFW Fish) in proximity to four geographically distinct marine reserves as part of hook-and-line monitoring activities (Fig. 1). Oregon Marine Reserves are fully protected areas meaning that ocean development and fishing are not allowed (*ODFW, 2022*). Charter samples were dissected in the field after being fileted, while the ODFW fish samples remained whole and frozen in plastic bags until dissection. Once in the lab, the ODFW fish were wiped with ethanol prior to dissection. Since we could not wipe the Charter samples with ethanol, we laid out a background contamination filter while dissecting the fish to address possible contamination in the field. The Charter samples were collected in summer 2018 and spring of 2019, while the ODFW samples were collected in the spring and fall of 2018 and 2019.

Overall, we processed a total of 58 subadult samples (GI tracts dissected from the whole fish) from across these areas near marine reserves (Cascade Head—10, Cape Perpetua—13, Cape Falcon—10, Redfish Rocks—11) and Newport, Oregon. GPS coordinates were noted for each fish collected by ODFW, showing the distance the fish were collected from a specific marine reserve. The average distance the fish were collected from their respective marine reserve ranged approx. between 6.8 km and 16.7 km. Although many Black

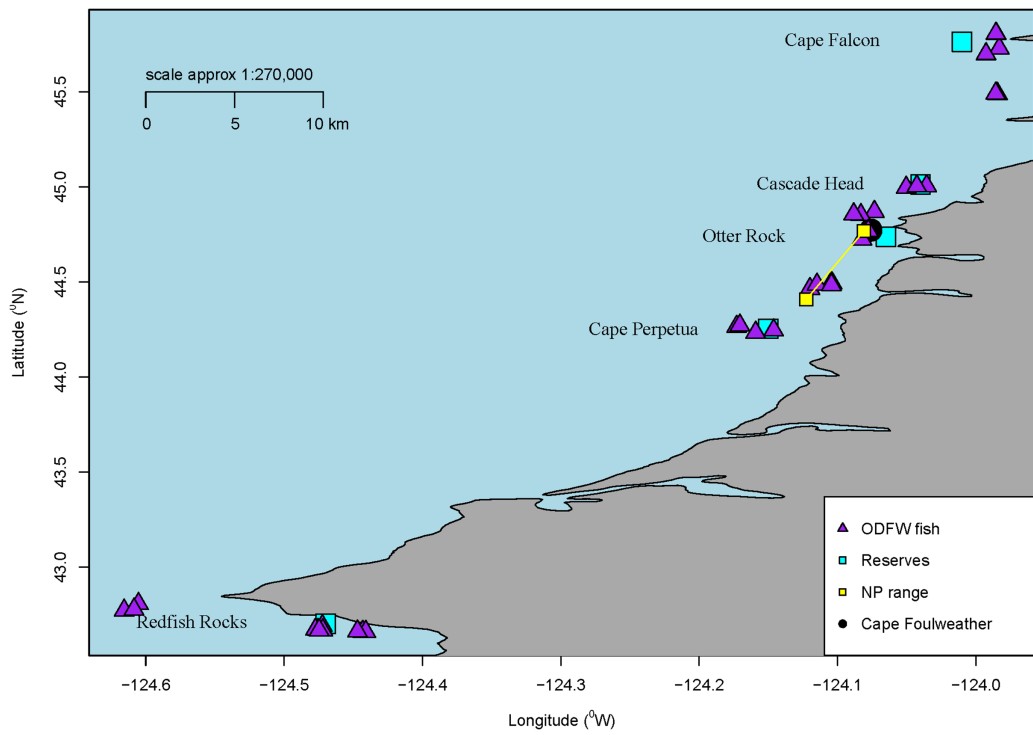

**Figure 1 Sample locations.** Map of all sampling locations for the fish collected by ODFW or the Charter company in Oregon, USA for microparticle ingestion. The marine reserve locations are all labelled along with a blue square. Additionally, all ODFW fish are shown on the map in triangles from GPS coordinates. Charter coordinates of the reserve fish. Newport fish are shown as a range. Finally, Cape Foulweather is shown with a black circle.             

Rockfish tend to have a high degree of site fidelity (*Green & Starr, 2011*), they have also been shown to move among sites (*Parker et al., 2007*), and therefore, we consider these sample fish to generally represent populations found in Oregon. ODFW does not need a collection permit for collections deemed relevant for research and management purposes.

## Pelagic juvenile rockfish collection

Juvenile samples were collected through a collaboration by Oregon State University, the Oregon Department of Fish and Wildlife's Marine Reserves Program, and the Oregon Coast Aquarium at two sites along the central Oregon Coast: Otter Rock, a marine reserve, and Cape Foulweather, a non-marine reserve site that is similar to the marine reserve in habitat, depths, oceanography, and historical fishing pressure (Fig. 1). While similar in habitat and topography, there is a difference in activities permitted between the reserve and comparison area. A comparison area is a separate location from the marine reserve, but in the vicinity of the reserve. These are areas where fishing and ocean development are allowed and also where monitoring occurs, while reserves prohibit fishing and ocean development. (*ODFW, 2022*). The fish were captured using Standardized Monitoring Units for the Recruitment of Fishes (SMURFs). Each SMURF consisted of low-density polyethylene (Fig. S2) mesh folded into a cylinder using garden fencing to form a structure similar to a kelp canopy, thus simulating natural structures which juveniles are known to

settle to (*Ottmann et al., 2018*). The samples were retrieved by snorkelers using a butterfly net to enclose the trap while it was brought on board the vessel. Juvenile fishes sorted from the SMURF were immediately euthanized using 2 mM tricaine methanesulfonate (MS-222) buffered in 6 mM sodium peroxide. The samples were identified to species and frozen for later laboratory use (*Ottmann et al., 2018*). Overall, 66 juvenile Black Rockfish samples were processed for microparticles by inspection. Juvenile samples were collected under National Marine Fisheries Service (NMFS) permit #18058, and juvenile fish protocols were approved by Oregon State University Animal Care and Use Protocol #4183.

## TERMINOLOGY

Microparticles, which include microplastics, are picked pieces that appear to be synthetic. We used four classifications to differentiate the materials of the verified particles, including: anthropogenic unknown, anthropogenic cellulosic, natural, and synthetic (true microplastics) and analyzed approximately 30% using uFTIR. For a particle to be classified as synthetic or natural, the correlational value matches had to be of synthetic or natural materials, respectively. 'Anthropogenic unknown' refers to verified particles with a mixture of readings of synthetic, natural, and/or cellulose matches. 'Anthropogenic cellulosic' refers to verified particles that were a mixture of cellulose. In the results section, all verified pieces that have been verified *via* FTIR, are referred to as 'spectra', 'pieces', 'verified particle' or the classifications listed above.

### Dissection and microplastic analysis

Dissection and microscopy procedures were conducted in a laminar flow hood (Erlab Captair Flow 391; Erlab USA, Rowley, MA, USA) (230 $m^3$/h processed air flow) with a HEPA filter or a fume hood with HEPA filter, using a Leica EZ4W scope with an attached camera or a Leica EZ4 dissecting microscope and added camera (Moticam 3.0 camera) at 8-35X magnification, without the additional camera zoom (Leica, Letzar, Germany). A fume hood was utilized only briefly while working with hazardous chemicals. Otherwise, a laminar flow hood with HEPA filtration was used the majority of the time to prevent microplastic contamination.

All fish (sub-adult and juveniles) were weighed pre-dissection except for the Charter fish as they were filleted before we dissected them. Subadult fish were used if they have a total length of 34–45 cm. The average weight of the subadult fish from ODFW were 1,044.1 g. We dissected the subadults and extracted and measured (length and weight) the GI tract from the esophagus to the vent. Additionally, undigested contents in the stomach were removed. Due to the juveniles' reduced size, all organs collected from the esophagus to the vent were digested for microplastic examination. We placed the GI tract of subadult fish into mason jars with the interior lining of the lids facing outwards, while juvenile guts were preserved in 20 ml glass vials. For subadult fish, we added 20% potassium hydroxide (KOH) at approximately three times the wet weight of the GI tract or 100 ml minimum. We digested juvenile samples in approximately 5 mL of the 20% KOH solution, which was roughly equivalent to the relative volume used for subadults. All samples were digested for 48–72 h in a water bath at 50 °C (either a Fisher Scientific Model Isotemp 220, Waltham,

MA, USA or a Precision Scientific 180 Series Water Bath, Chicago, IL, USA), although some of the subadult samples were able to continue digesting at room temperature after the fact.

We sieved all subadult samples through stacked 1 mm (first) and 63 μm sieves (juveniles were not sieved due to their size) and then vacuum filtered the remaining liquid by pouring the solution onto a 5 μm Whatman polycarbonate filter using a Büchner funnel. A separatory funnel was used for most samples to ensure liquid was being poured in the center of the filter. Any large item on the sieves were looked at under a scope, and everything else was vacuum filtered. For the subadult fish, we filtered 100–300 mL per filter. Alcojet was used as needed to break apart lipid heavy digestate (*Crichton et al., 2017*). As juvenile fish samples were stored in glass jars, the jars and caps were rinsed and the resulting RO water poured onto the vacuum filter as well, resulting in 5 to 25 mL of filtered volume per sample per filter. We imaged all suspected plastics using a Leica EZ4W (camera included) or Leica EZ4 microscope equipped with a Motic 3.0 MP camera based on guidelines from *Rochman et al. (2019)* and *Lusher et al. (2020)*. All but one of the microparticles that were run with FTIR were measured using the Leica EZ (mm), LevenhukLite software (mm), or "Motic Images Plus 3.0 ML" (μm), to ensure they were in the microparticle size range (<5 mm in one direction). The microparticles from juveniles were measured by one author, and the microparticles from the subadult fish were measured by another. All microparticles from subadults were measured in mm but converted to um. Despite the accidental exclusion of this single piece from being measured, it was retained in the results.

We followed protocols from *Rochman et al. (2019)* to determine morphology and color of the microparticles during this process as accurately as possible. Most of the filters were picked for microparticles for 1–3 h on average for the subadult fish at a later date. Once picked, suspected plastics (*e.g.*, anthropogenic cellulosic, natural, synthetic, anthropogenic unknown) were placed into a verified polystyrene (marketed as acrylic) container or onto a double-sided piece of tape on a piece of projection article. We visually inspected these to determine if they could be plastic by probing them to see if they broke. If they broke, we did not count them as potential microparticles for future FTIR analysis (*Lusher et al., 2020*), unless they had already been verified *via* FTIR. There were some instances where we picked a subset of the microparticles found on filters due to the similarities of the pieces or picked pieces under different light. The term "microparticle" along with the classification categories described below are used to help differentiate microparticles from those that are analytically confirmed to be microplastics (*Miller et al., 2021*; *Harris et al., 2021a*).

## Fourier transform infrared spectroscopy

We analytically verified a subset (~30% of all microparticles) of all suspected plastics from fish and blanks (~30% per *Brander et al., 2020*) on a Thermo Fisher Nicolet iN5, smart iTX, and Nicolet iS20 FTIR (Waltham, MA, USA). The particles were subsampled to represent the most common particle types collected. All microparticles were measured with Attenuated Total Reflectance or reflectance, and then followed by analysis using micro-Attenuated Total Reflectance (μATR). OMNIC software was used to generate

spectra and Open Specy was used to provide final identification of material type. ATR with a Diamond or Germanium tip was used to run scans (64–256 in subadult fish, 128 in juveniles) of each microparticle to generate the resulting spectrum. We atmospheric corrected each spectrum no matter when the background was collected. This was due to the location of the FTIR relative to air sources. We matched spectra using several libraries including a library from *Primpke et al. (2018)*, one created by the NOAA Marine Debris Research Program at UNCW, libraries created in-house, as well as standards that came with the machine. However, we did not match all pieces against all libraries as libraries were formed as this study progressed.

When a spectrum is matched to a library in OMNIC, a correlational match is produced. Due to the low qualifying spectra obtained for some spectra in OMNIC, the spectra generated were subsequently verified using Open Specy (*Cowger et al., 2021*). OMNIC provides the top 10 correlational matches for each spectrum, while Open Specy provides 100, and Open Specy uses multiple libraries specific to plastic polymers. To accept the generated spectrum for this study in Open Specy, the top five matches had to be >70. For consistency, all microparticles analyzed through Open Specy were smoothed at level 3, with baseline corrections at level 8. We did not account for the Ge tip minimum at $675 \text{ cm}^{-1}$. We categorized the microparticles into four categories based on Open Specy results: anthropogenic unknown, anthropogenic cellulosic, natural, and synthetic based on characterizations from *Miller et al. (2021)* similar to: *Harris et al. (2021a)*, *Caldwell et al. (2022)*. However, the microparticle holding jar and net that were verified with FTIR were not checked with Open Specy.

## Quality assurance

We used several control procedures to account for background contamination. We cleaned all glassware using soap and water, rinsed with DI and RO water, then rinsed with 70% ethanol, and wrapped in foil prior to being baked at 400 °C for 4 h. We stamped a Whatmann filter with a $12 \times 12$ box grid and placed it into a glass petri dish each day a fish sample was open under the hood. This allowed us to account for plastic pieces that potentially deposited from the air into our samples within the hood. Our laminar flow hood was set up to allow computer access outside of the sample workspace, to reduce the potential for contamination.

We employed several blanks throughout the different microparticle processes. We used filtered and unfiltered DI, Milli-Q, and RO water (water source was switched mid-project due to lab move) to make solutions and to wet filters. To account for any contamination from water, we collected and vacuumed samples following the above procedures. We also used air blanks and KOH procedural blanks to track contamination throughout the process; however, we only included a subset in the final analysis. Finally, we followed recommendations to ensure minimal background contamination (*Brander et al., 2020*; *Cowger et al., 2020*). This included all project personnel wearing 100% cotton lab coats, mostly 100% cotton face masks (due to the COVID-19 pandemic), clothing as close to 100% cotton as possible, and nitrile gloves. Additionally, the container used to hold some
microparticles and the mesh nets used for catching the juveniles were verified *via* µFTIR using 128 scans and a Germanium crystal (Fig. S2).

To determine the average number of microparticles attributed to background contamination (air, collection gear, personal accessories), the number of microparticles in the gastrointestinal tract was divided by the number of microparticles found in controls or blanks (air, KOH, water) and then multiplied by the number of controls divided by the number of fish (see Supplemental data), resulting in the number of microparticles per fish.

## Study limitations

Differences in microparticle counts between fish size classes could be due to the fish size difference itself, but it is important to note that ingestion studies typically provide a snapshot view of microparticle consumption. Differences in the timing of last feeding, relative abundance of microparticles when feeding, clearance rates of microplastics from the gut, and even relative fish abundance (*i.e.*, potential competition for prey items) could potentially affect instantaneous views of microparticles in the digestive tracts of fish. *Athey et al. (2020b)* found that polyethylene microspheres were retained within larval digestive tracts of silversides (*Menidia beryllinia*) up to 72 h after feeding, but we do not know the timing of passage for sub-adult and juvenile Black Rockfish. Given that the digestive period is relatively brief in most fish species, we have likely underestimated microparticles including microplastic consumption in Black Rockfish.

Our study, like many others (*i.e.*, *Jamieson et al., 2019*; *Ibrahim et al., 2020*; *Li et al., 2020a*; *Ragusa et al., 2021*; *Caldwell et al., 2022*), did not include a recovery experiment. Recovery studies should be completed for experiments in the future to understand the success of the methods used. Additionally, this test could have provided beneficial information for understanding whether the concentration of KOH and temperature used in the oven degraded particles. Finally, our study is highly dependent on microscopy, as it is one of the first steps in particle recovery. Our FTIR is a manual machine, and therefore we needed to pick each individual particle that was a microparticle. *Kotar et al. (2022)* shows that microscopy is not the most effective method for microplastic analysis. With this in mind, we most likely underestimated the number of particles, especially as they decreased in size.

## Data analysis

We conducted all statistical analyses in the R software environment using several libraries (car, carData, CARS, ggplot, multcomp, cowplot, mapdata, mapplots, maptools) and Excel. To determine if there was a statistically significant difference between the sampling sites and to account for differences in sample sizes, we conducted a generalized linear model (GLM) with Poisson error and log link using a GLM with Poisson error and a Tukey *post-hoc* test to identify any differences in the total number of pieces per fish across sites, at an alpha of 0.05. Wilson's Score Intervals were used to find intervals for fish containing microparticles by site (Table S1). Additionally, confidence intervals can be found in Tables S4 and S5. Raw data and code can be found at: https://github.com/Brander-Harper/Black-Rockfish-Manuscript.
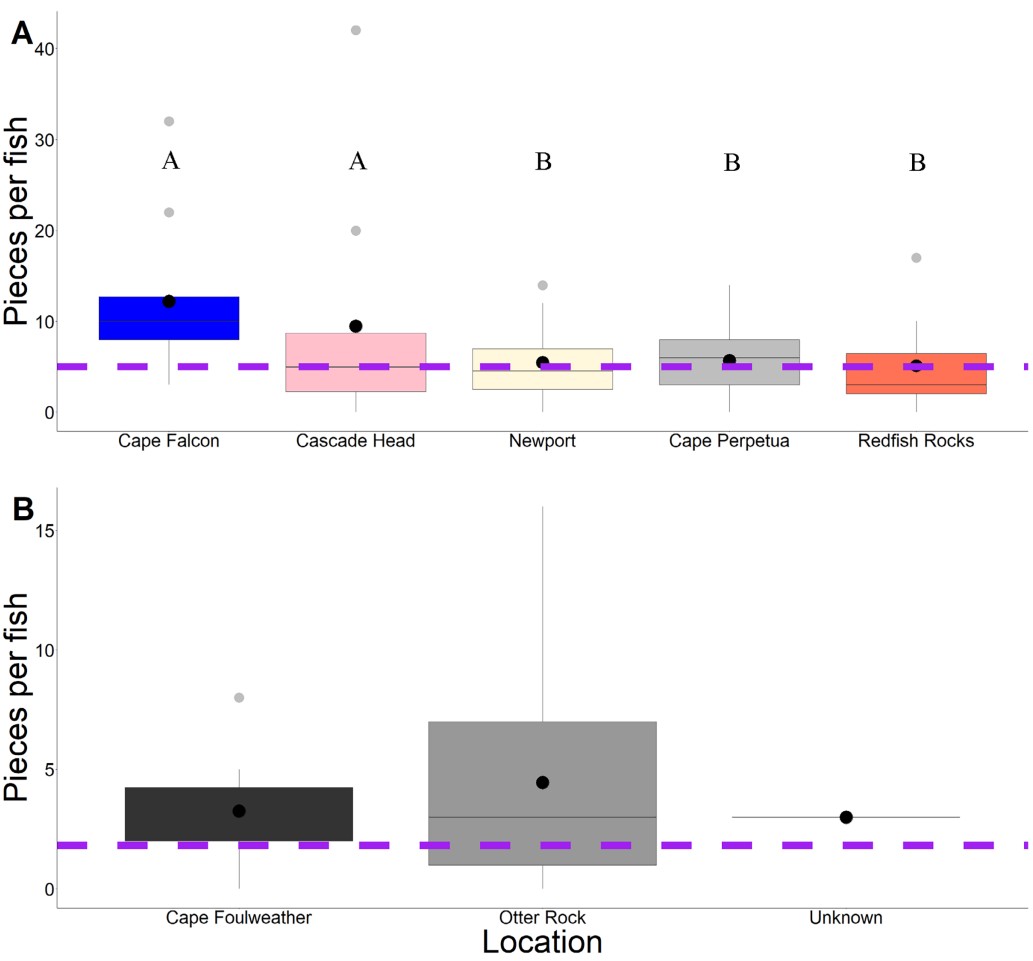

**Figure 2 Microparticles per fish by site.** Microparticles from the subadult (A) and juveniles (B). Different capital letters represent statistical significance ($P < 0.05$) (ANOVA, Tukey *post-hoc*, Poisson distribution). Statistics were completed individually for the subadults and the juveniles. Mean values are represented by black circles, while the grey circles represent outlying points. The boxplot represents the first and third quartile, median, and whiskers. The dashed line represents the potential background contamination for both the subadult and juvenile fish. The fish collected from Cape Falcon, Cascade Head, Cape Perpetua, Redfish Rocks, and Cape Foulweather were all collected in the vicinity of marine reserves, while the fish collected in Newport (Charter) were not.

## RESULTS

Across 58 subadult fish (34–45 cm, total length and 1,044.1 g average weight for ODFW and Charter fish) from the vicinity of four marine reserves and Newport, Oregon, and 66 juvenile fish (3.49 cm–5.96 cm, TL) from Otter Rock Marine Reserve and Cape Foulweather, we found 424 and 278 microparticles, respectively. Therefore, on average, 7.31 (average background = 5) microparticles per subadult fish and 4.21 (average background = 1.8) microparticles per juvenile fish were detected (Fig. 2). Geographically, there was no significant difference (GLM, Tukey *post-hoc*, poisson distribution; Table S2, confidence intervals; Table S4) in the presence of microparticles in subadult fish between Cascade Head and Cape Falcon ($P > 0.05$), or among Newport, Cape Perpetua, and Redfish

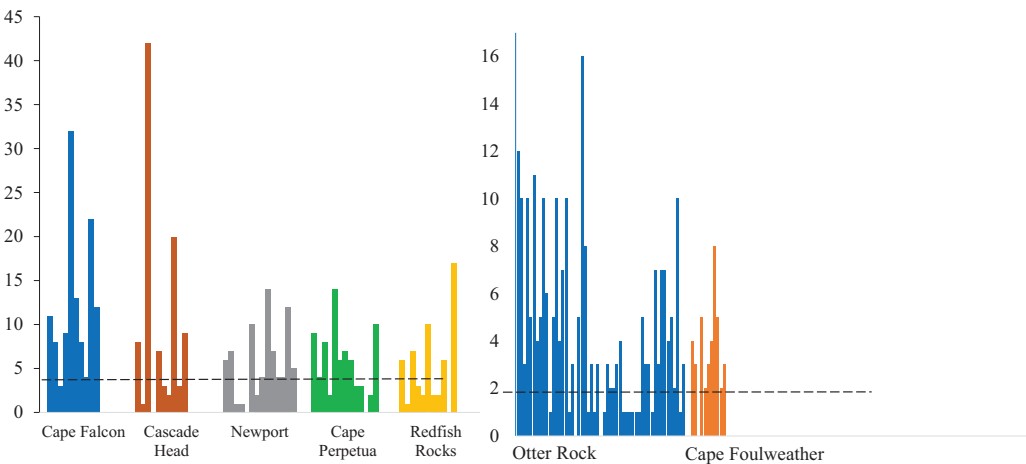

**Figure 3 Microparticles detected in each gastrointestinal tracts.** The number of microparticles per fish by site for subadult fish (left) and juvenile fish (right). Vertical lines represent the number of microparticles found in each fish. The dashed line for subadult fish (y = 5) and juvenile fish (y = 1.82) is the estimated average number of background pieces per fish. One fish from an unknown location has been removed. CH refers to Cascade Head, CP refers to Cape Perpetua, CF is Cape Falcon and RR is Redfish Rocks.

Rocks. There were significant differences ($P < 0.05$) between the northern most sites (Cascade Head and Cape Falcon) and the remaining more southern sites (Newport, Cape Perpetua, and Redfish Rocks) (Fig. 2).

In contrast, there was no significant difference ($P > 0.05$) in microparticle presence in juvenile fish from the Otter Rock Marine Reserve and the comparison site at Cape Foulweather (Table S3, confidence intervals; Table S5). The average number of background pieces per fish was 5 pieces for subadult fish and 1.8 pieces for juveniles (Fig. 3).

We classified microparticles by color and morphology prior to FTIR analyses. We found 11 and seven microparticle colors in the subadult and juvenile fish respectively (Fig. S1). Of the 13 size classes used for the length of microparticles found in the juvenile and subadult fish, 10 size classes were present for the subadult and 11 for juveniles (Fig. 4). The 500–1,000 µm size class was the most common size class of all microparticles for both groups of fish (Fig. 4). The longest verified fiber from the juveniles was 5,693.96 µm, while the longest verified fiber for the subadults was 3,113.3 µm (Fig. 4). Even with the juveniles appearing to consume longer microparticles than the subadults, only the verified microparticles were measured, therefore, there could be a wider range of lengths in the microparticles that was not verified. There were six different morphologies of microparticles in the subadult fish whereas the juveniles had four microparticle morphologies. In both sets of fish, fibers were the most common morphology (Fig. 5).

Of the 424 microparticles picked from the subadult fish, a subset of 131 passed the necessary threshold for use in this study (top 10 matches in OMNIC >40 and top 5 matches in Open Specy >70) (Fig. 6). However, over 43% of the total samples (>180 microparticles) were run on FTIR to have enough qualifying spectra (OMNIC values >40 and top five matches in Open Specy >70). Of the 131 qualifying spectra, 16 (or 12.21%)

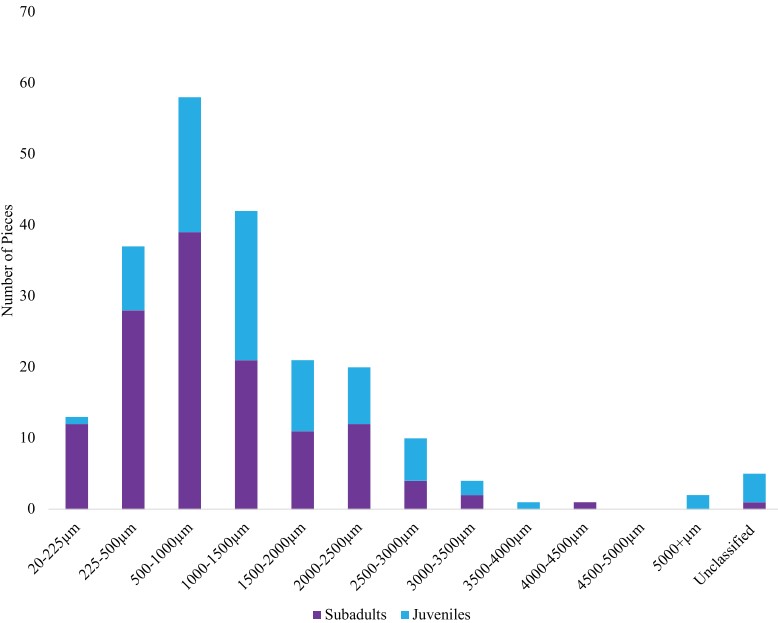

**Figure 4 Size distribution of verified microparticles of the subadult and juvenile fish.** Distribution of verified microplastics by FTIR for juveniles (blue) and subadults (purple).

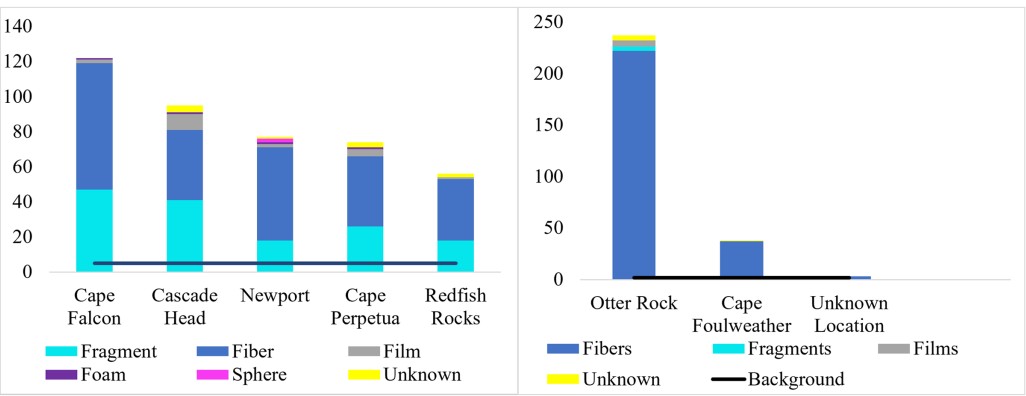

**Figure 5 Types of microparticle morphologies by site.** Morphological breakdown of microparticles found in the subadult (right, *n* = 58) and pelagic juvenile (left, *n* = 66) Black Rockfish collected from the Oregon coast.

were synthetic (or true microplastics) (Fig. 6). Another 72 pieces were classified as anthropogenic cellulosic (54.96%), 42 pieces were anthropogenic unknown (32.06%), and one piece was natural (0.76%) (Fig. 6).

Of the 278 microparticles picked from the juveniles, a subset of 83 pieces were run and had qualifying spectra. Of these, 10 were synthetic (or true microplastics) (12.05%) (Fig. 6). Three of the 83 (0.04%) were anthropogenic unknown followed closely by 68 anthropogenic cellulosic (81.93%), and finally two pieces (2.41%) were classified as natural (Fig. 6). Tables S6 and S7 show the OMNIC and Open Specy classifications for each particle presented that was verified *via* FTIR.

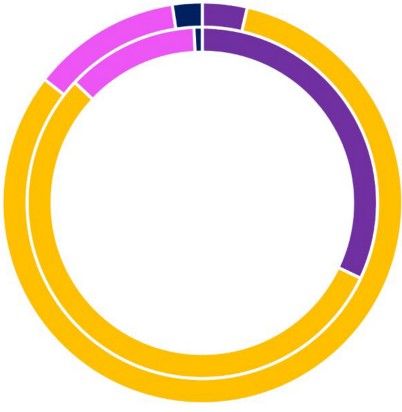

■ Anthropogenic unknown  ■ Anthropogenic cellulosic  ■ Synthetic  ■ Natural

**Figure 6 Classifications of verified microparticles by percentage for subadults and juveniles.** Classification percentages of particles verified *via* uFTIR (OMNIC, Open Specy) for subadults (interior ring) and juveniles (exterior ring) collected off the Oregon coast. Approximately 30% of all microparticles from the fish were analyzed for the adults (*n* = 131) and juveniles (*n* = 83).

Of those analyzed *via* µFTIR, all synthetic pieces (*n* = 10) in the juveniles were fibers, while 52.9% (*n* = 9) of the synthetic pieces in subadult fish were fibers. The remaining synthetic pieces from the subadult fish were films (*n* = 2), fragments (*n* = 4), and foam (*n* = 1). For the subadult fish, the synthetic pieces consisted of a mix of different plastic types including but not limited to cellulose propionate, acrylonitrile butadiene styrene, paint, polyurethane, PET/polyester, resins, and polyethylene. For the juvenile fish, eight of the 10 synthetic pieces were composed primarily of polyester or of a polyester derivative (80%) (polyesterpthalate, polyesterterapthalate) followed by equal parts HDPE (10%) and polypropylene (10%). The 16 synthetic microplastic pieces subsampled from and analyzed in subadult fish were composed of eight colors. Black (*n* = 4) was the most common synthetic color found, followed by blue (*n* = 3) (Fig. 7). For the juveniles, the 10 synthetic microplastic pieces were composed of three colors: blue (*n* = 6), clear (*n* = 3), and red (*n* = 1, Fig. 7).

## DISCUSSION

Documenting that subadult Black Rockfish are regularly ingesting microparticles and microplastics across life stages and spatial scales contributes to our knowledge of plastic pollution impacts along the Oregon coast. It is notable that, despite the size disparities, the juvenile fish consumed similar percentages of microplastics to that of the subadult fish (12.21% *vs.* 12.05%), which could be due to similar distributions of microparticles along much of the Oregon coast, which need to be identified (*Horn, Granek & Steele, 2020*). It is important to note that these samples are a snapshot in time and therefore are likely an underestimation of what is being internalized, as has been acknowledged in other studies (*Wright, Thompson & Galloway, 2013*; *Baechler et al., 2020a*). While we studied two different life stages, Black Rockfish are opportunistic feeders and may be predating upon microparticles or obtaining them from prey in a similar manner at both the juvenile and

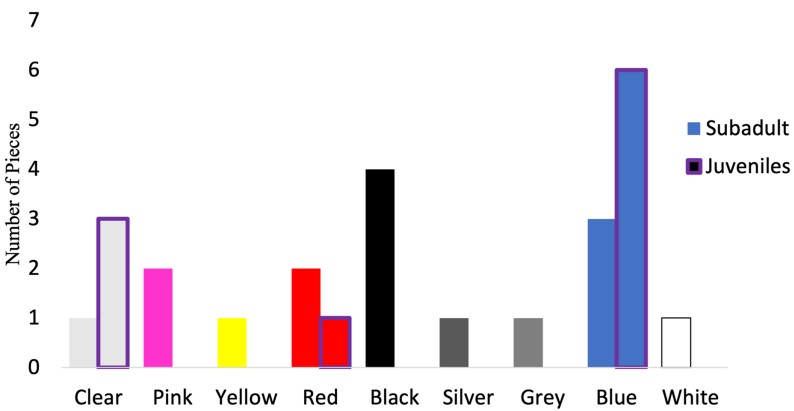

**Figure 7 Colors of verified microparticles for the subadults and juveniles.** Numbers of verified pieces of synthetic material by colors Juveniles (purple highlight, $n = 10$) and sub-adult larger fish (solid bars, $n = 16$). The hatched bars start on top of the solid bars.

subadult stages. Our result, that ~12% of microparticles are synthetic in Black Rockfish, is similar to the 9.5% of Stardum sp. (*Stellifer brasiliensis*) GI tracts found by *Amorim, Ramos & Nogueira Júnior, (2020)* in Brazil, but somewhat lower than what other researchers have found for plastics presence in fishes. For example, *Barboza et al. (2020)* found plastics in 49% of the fish they examined with ATR-FTIR from the Atlantic Ocean (NE), while *Alomar et al. (2017)* found plastics in approximately 27% of their fish by using imaging FTIR from the Island of Mallorca. However, our results were somewhat higher than the results found in *Covernton et al. (2022)* completed in British Columbia, Canada. The authors in this article found that Black Rockfish only contained a mean concentration of 1.12 (0.52-2.13) in digestive tracts with full stomachs or 0.10 (0.01-0.44) when their stomachs were empty (*Covernton et al., 2022*).

Fiber-like microparticles are one of the most commonly detected morphologies (*Rochman et al., 2015*; *Baechler et al., 2020a*; *Barboza et al., 2020*; *Amorim, Ramos & Nogueira Júnior, 2020*; *Harris et al., 2021a*). Just like all microplastics, fibers themselves are complex (*Rochman et al., 2019*; *Athey & Erdle, 2021*). Fibers can be classified several ways including natural, semisynthetic and synthetic (*Granek et al., 2022*; *Athey & Erdle, 2021*); therefore, it is important to look at all fiber materials as they are commonly found (*Athey & Erdle, 2021*). In alignment with this, fibers were the most common morphology in both the subadult fish and juveniles. Beyond morphology, black was the most common color of microplastics we found in our subadult fish and blue the most common color of microplastics in juveniles. These results are consistent with many studies looking at verified microplastics. Blue was the most common color of microplastics found in several fish species including Atlantic Horse Mackerel (*Trachurus trachurus*), European Seabass (*Dientrachus labrax*), Atlantic Chub Mackerel (*Scomber colias*), and a species of Stardrum (*Stellifer brasiliensis*) (*Barboza et al., 2020*; *Amorim, Ramos & Nogueira Júnior, 2020*). Microplastics found in Striped Red Mullet (*Mullus surmuletus*) had blue and black as the most common colors (*Alomar et al., 2017*). Although we cannot be certain as to why blue fibers are commonly detected, possible sources are blue jean fibers from wastewater

treatment plant effluent (*Athey et al., 2020a*) or fishing rope (*Xue et al., 2020*). The size distribution of the verified microparticles found in the juvenile fish was 167.18-5,693.96 μm, while it was 81.167- 4,480 μm for the subadult fish. Our results are similar to other studies in that researchers find a wide range of microparticle sizes being internalized by fish (*Arias et al., 2019*; *Amorim, Ramos & Nogueira Júnior, 2020*). Similar to our study, *Arias et al. (2019)* found microparticles ranging up to 5 mm and up to 4.7 mm in *Amorim, Ramos & Nogueira Júnior, (2020)*. Despite how small the juvenile fish in this study were, *Steer et al. (2017)* found microparticles ingested by larval fish ranging from 50 to 1,100 μm. Although we don't know what the juveniles consumed prior to being caught, we could expect that prey size or what the prey ate could impact the particle sizes seen in our study. By considering the predominant morphologies and colors of microplastics and microparticles being consumed by these fish, it aids in our understanding of the role of microparticles in Oregon waters.

## Impacts of anthropogenic pieces

Microparticles across all types were consumed by Black Rockfish, at an average of 7.31 (average background = 5) microparticles per subadult fish and an average of 4.21 (average background = 1.8) microparticles per fish for the juveniles (5.56 for all fish), with the majority (86.4%) of the verified polymer types being of unknown origin or cellulose based. A potential source of some of these anthropogenic or natural fibers could be from rope in the Dungeness fishery in Oregon. The material used for the twine on the pots for the largest commercial fishery in the state is cotton (*Oregon Deparment of Fish & Wildlife, 2021*). Our results are similar to *Rochman et al. (2015)*, in which anthropogenic (albeit unverified) pieces were more prevalent than plastic in fish from the USA. Our results align with their finding that 12 study species were exposed to more textile fragments than synthetics, including Pacific Oysters (*Crassostrea gigas*), Yellowtail Rockfish (*Sebastes flavidus*), Chinook Salmon (*Oncorhynchus tshawytscha*), Vermilion Rockfish (*Sebastes miniatus*), Blue Rockfish (*Sebastes mystinus*), Pacific Mackerel (*Scomber japonicus*), Striped Bass (*Morone saxatilis*), Copper Rockfish (*Sebastes caurinus*), Lingcod (*Ophiodon elongatus*), Jacksmelt (*Atherinopsis californiensis*), Albacore Tuna (*Thunnus alalunga*), Pacific Anchovy (*Engraulis mordax*), and Pacific sanddab (*Citharichthys sordidus*). Due to the recurring presence of anthropogenic pieces outside of plastic polymers, a better understanding of their impacts on the health of organisms and humans is needed to assess their potential risk. Twenty-one Black Rockfish contained verified synthetics in our study, which is 17%; however, in *Amorim, Ramos & Nogueira Júnior (2020)* 12.4% of the fish contained plastics, but these pieces were not verified. Ingestion in our study is much higher than the 0.03% of fish that contained plastics examined by *Steer et al. (2017)* and similar to results found by *Arias et al. (2019)*. *Arias et al. (2019)*, showed that there were 241 non-verified microplastics found in 20 (12.05%) fish. Standard guidelines and methods are necessary to improve reproducibility and the ability to trace microparticles back to their source (*Cowger et al., 2020*).

Due to the complexities involved in verifying microparticle composition, some studies do not verify the material type using analytical verification (*Rochman et al., 2015*; *Arias*

*et al., 2019*; *Amorim, Ramos & Nogueira Júnior, 2020*), but many studies such as ours now do (*i.e.*, µFTIR). However, it is important to that note microparticles that are semi-synthetic or natural still need to be examined, as they are frequently detected (*Athey & Erdle, 2021*) and analytical identification is needed (*Rochman et al., 2015*; *Steer et al., 2017*; *Alomar et al., 2017*; *Rochman et al., 2019*; *Allen et al., 2019*; *Brander et al., 2020*; *Cowger et al., 2020*) to differentiate natural and cellulosic materials from synthetic or anthropogenically modified fibers, as they can also be quite similar visually (*Athey & Erdle, 2021*). There are also several different definitions that are used to classify microparticles that are neither plastic nor natural. Some of these include anthropogenic cellulosic, unknown potentially rubber, anthropogenic unknown, and unknown (*Miller et al., 2021*). Like the current lack of standardized analytical methodology, there is currently no standard terminology in use for microparticles that are not either clearly plastic or natural. Due to these complexities, we used both OMNIC (Thermo Fisher) and Open Specy (*Cowger et al., 2021*) to verify our spectral matches. Future analysis using FTIR could be improved upon by splitting the microparticles when possible before analysis to expose un-weathered or rugose surfaces, as this can produce increased FTIR matches (*Jung et al., 2018*). While the microplastics field has matured, the continued lack of standardization suggests the need for better harmonized methodologies for easier identification of microplastics and improved consistency (*Brander et al., 2020*; *Cowger et al., 2020*).

We sought to standardize methods across all of our analyses, and there were differences in the number of background microparticles detected for the different fish size classes; subadult fish had an average of five while juveniles had an average of 1.82. This could be due to the timing of when the samples were processed, how samples were stored and collected, or that the samples were processed in two different hood configurations. Regardless, this emphasizes the importance of these controls in the interpretation of microparticle data.

## Toxicology

We did not measure the toxicity of the plastics in our fish. However, a number of studies have shown that reproduction is impacted when fish are exposed to different types of plastics at different sizes and concentrations (*e.g.*, *Chisada, Yoshida & Karita, 2019*; *Zhu et al., 2020*), though some have not (*Jacob et al., 2020*). Additionally, plastics have been shown to cause cellular changes in fishes (*e.g.*, (*Karami et al., 2016*; *Jabeen et al., 2018*; *Espinosa, Esteban & Cuesta, 2019*; *Ahrendt et al., 2020*; *Jacob et al., 2020*)) Fibers, specifically, have been used in toxicology studies to determine the impact of internalization and are known to impact body condition and to cause cellular changes in goldfish (*Carassius auratus*) (*Jabeen et al., 2018*) and respiration impacts in Black Sea Bass (*Centropristis striata*) (*Stienbarger et al., 2021*). Additionally, *Amorim, Ramos & Nogueira Júnior (2020)*, saw that fibers, along with other morphologies, decreased Fulton's K in subadults of *Stellifer brasiliensis* and hepatosomatic index in *Micropogonias furnieri* (*Arias et al., 2019*). Microplastics have been shown to affect a variety of endpoints in marine organisms, therefore, it is suspected that they can affect population stability. However,

additional studies are needed to better understand the impact of microplastics on both population stability and ecosystem stability (*Baechler et al., 2020b*).

Fibers or filaments have been detected in several human samples including, but not limited to, colectomy (*Ibrahim et al., 2020*) and lung tissue (*Jenner et al., 2022*), however in a study on stool samples, fibers were not commonly found (*Schwabl et al., 2019*). Additional studies have looked at plastics in humans but did not look at morphologies and/or find fibers (*e.g.*, (*Ragusa et al., 2021*; *Leslie et al., 2022*)).

Like many labs, we could not detect particles <20 um. Most labs have difficulty with this as seen in a multi-laboratory comparison study led by the Southern California Coastal Water Research Project (*De Frond et al., 2022*). In a article written by *De Frond et al. (2022)*, the authors explain that the participants in the study (including our lab) had a recovery for particles >20 um at 92%. However, when considering particles <20 um, it is 32%. Since we do not have the ability to collect and then verify pieces that are this small, we risk missing many micro- and nanoparticles that can be consumed by marine life and potentially cause toxic effects (*e.g.*, *Cunningham et al., 2022*; *Siddiqui et al., 2022*).

## Spatial distribution

There are significant differences by location for microplastic ingestion by subadult Black Rockfish. ODFW samples from the two northern-most areas, near Cape Falcon and Cascade Head, had significantly higher levels of microparticle ingestion in comparison to those caught near Newport, Cape Perpetua and Redfish Rocks (Table S2). The lack of microparticle ingestion difference by juvenile Black Rockfish between Otter Rock marine reserve and Cape Foulweather is not surprising, as these two locations are physically close to one another (~3 nautical miles) (*Oregon Department of Fish & Wildlife, Marine Resources Program, 2014*). Although more data are needed to determine why there was a statistical difference in microparticle ingestion among sites, one explanation could be the northernmost sampling sites are in closer proximity to the Columbia River. Rivers can concentrate plastics across a broad terrestrial region that include dense human population centers away from the coast (*Harris et al., 2021b*), and the Columbia River, which flows though many municipalities including the Portland Metropolitan area, is, known to be contaminated with microplastics and other particle types but predominantly fibers (*Talbot et al., 2022*).

Our work suggests that fish located inside the reserves contain a similar amount of microparticles in both juvenile and subadult Black Rockfish, although we only compared juvenile Black Rockfish inside a comparison area to one marine reserve (Otter Rock, *Oregon Department of Fish & Wildlife, Marine Resources Program (2021)*). While ODFW subadult fish samples were not caught inside the reserves, the oceanographic processes known to influence comparison areas where samples were taken and reserves are similar (*ODFW, 2022*), suggesting similar microparticle exposure levels should be expected. This assumes that exposure to microparticles occurs mainly due to oceanographic processes and not from local land-based, inputs. Finally, it is of particular interest that there are several reasons to create and maintain marine reserves (*White et al., 2010*; *Abessa et al., 2018*) and many do not include protection from pollution, as was the case in Oregon.

In many cases marine reserves have been placed in areas that are pre-contaminated with chemicals (*Abessa et al., 2018*) or are unknowingly in the pathway of newly emerging contaminants such as microplastics and other microparticles. Future marine reserve or protected area planning may benefit from considering locations of potential microplastic sources. This will enable careful consideration of reserve location, especially in coastal areas with both terrestrial and aquatic sources of plastics.

## CONCLUSIONS

Despite the rapid advancement of the microplastics field, there remain knowledge gaps regarding environmental relevance and microparticle concentrations. There does not currently seem to be a consensus as to what concentration or number of plastics are biologically relevant. Once this is determined, we can better design studies to learn how organisms will respond to microparticulate exposure under additional stressors and in different parts of the world.

Microparticles including microplastics were consistently detected in the gastrointestinal tracts of both juvenile and subadult Black Rockfish sampled along the Oregon coast and in the vicinity of marine reserves. These fish, collected from several locations spanning ~179 nautical miles (North to South), demonstrated that plastic is indeed found in nearshore waters of the northeast Pacific. Although we do not know the exact sources of these microplastics, verification that they are indeed synthetic or of other origin can help guide future research and inform risk assessment. Future research and more standardized methodologies are needed to help quantify the level of microplastic ingestion that might be associated with adverse responses in fish.

## ACKNOWLEDGEMENTS

We wish to acknowledge Emily Pedersen, Jennifer Van Brocklin, Jordan Landry, Quan Luong, Dakota Fee, Kiera McNeely, Amy Pumputis, and Eli Spicer for their help in collecting and processing of the fish samples, figure preparation, and reading the article. We would also like to thank Sara Hutton, Anna Bolm and Núria Viladrich for help with R studio and providing code. We would also like to thank Will White for assistance with statistics. We thank the more than 60 individuals who assisted with the field collection of the juvenile fish samples and the Oregon Department of Fish and Wildlife, and Newport Marina Store and Charters, for subadult samples.

### Funding

This work was supported by The Agricutlural Reserach Foundation at Oregon State University (grant to SMB), Oregon State University research and teaching assistantships, and The National Science Foundation Research Traineeship project at Oregon State University (Award #: 1545188, Award Title: NRT-DESE: Risk and uncertainty quantification in marine science and policy). The funders had no role in study design, data collection and analysis, decision to publish, or preparation of the manuscript.

## Grant Disclosures

The following grant information was disclosed by the authors:

Agricutlural Reserach Foundation at Oregon State University.

Oregon State University Research and Teaching Assistantships.

National Science Foundation Research Traineeship at Oregon State University.

## Competing Interests

Susanne M. Brander is an Academic Editor for PeerJ.

## Author Contributions

- Katherine S. Lasdin conceived and designed the experiments, performed the experiments, analyzed the data, prepared figures and/or tables, authored or reviewed drafts of the article, and approved the final draft.
- Madison Arnold performed the experiments, analyzed the data, prepared figures and/or tables, authored or reviewed drafts of the article, and approved the final draft.
- Anika Agrawal performed the experiments, analyzed the data, prepared figures and/or tables, authored or reviewed drafts of the article, and approved the final draft.
- H. William Fennie performed the experiments, authored or reviewed drafts of the article, and approved the final draft.
- Kirsten Grorud-Colvert conceived and designed the experiments, authored or reviewed drafts of the article, and approved the final draft.
- Su Sponaugle conceived and designed the experiments, authored or reviewed drafts of the article, and approved the final draft.
- Lindsay Aylesworth conceived and designed the experiments, authored or reviewed drafts of the article, and approved the final draft.
- Scott Heppell conceived and designed the experiments, authored or reviewed drafts of the article, and approved the final draft.
- Susanne M. Brander conceived and designed the experiments, performed the experiments, analyzed the data, prepared figures and/or tables, authored or reviewed drafts of the article, and approved the final draft.

## Animal Ethics

The following information was supplied relating to ethical approvals (*i.e.*, approving body and any reference numbers):

Oregon State University Animal Care and Use Protocol approved the protocols used for the juveniles(#4183).

## Field Study Permissions

The following information was supplied relating to field study approvals (*i.e.*, approving body and any reference numbers):

Additionally, National Marine Fisheries Service granted a permit to collected the juveniles(#18058).

## Data Availability

The supplemental figures/tables, code and raw data are available at GitHub: https://github.com/Brander-Harper/Black-Rockfish-Manuscript.

## Supplemental Information

Supplemental information for this article can be found online at http://dx.doi.org/10.7717/peerj.14564#supplemental-information.

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
