# Peer review of "Presence of microplastics and microparticles in Oregon Black Rockfish sampled near marine reserve areas"

_PeerJ, doi:10.7717/peerj.14564_

## Round 0.1 · original submission · Major Revisions

· Academic Editor

Major Revisions

First, let me apologize for the length of time your manuscript has been in review.

The reviewer comments are relevant and generally straightforward. One reviewer in particular has asked that you pay special attention to the statistics used in your analysis. Also consider stating some hypotheses about microplastic ingestion by the rockfish at their different life stages due to differences in their habitat and feeding habits. Please respond to these and the other comments and revise your manuscript accordingly.

Some important points to consider center around the measurements of microplastics. Please consider reporting more details of the blank results and a discussion of the size aspect of microparticles including an evaluation of detection limits and measurement uncertainties.

Reviewer 1 ·

Basic reporting

This manuscript reports anthropogenic particle and microplastic concentrations in the digestive tracts of black rockfish. Overall, the writing is satisfactory and mostly clear, but could benefit from some editing for conciseness and clarity, especially in certain areas.

The article sufficiently references the relevant literature and provides enough background/context.

The article structure is mostly good, although I have some comments on this in the general comments section.

The article is generally self-contained. The authors do not exactly provide testable hypotheses, but this is also more of an observational study. It would, however, have strengthened the paper if they provided hypotheses about microplastic ingestion by the rockfish at their different life stages due to differences in their habitat and feeding habits, and then addressed these.

Experimental design

The research is original.

The research question is not very well defined, and this could be laid out better. It seems to just be "how many microplastic particles are in rockfish?"

The investigation is rigorous, and the authors use up-to-date methodology, although some reporting needs to be improved. Some of the methodology is not clearly described and should be clarified. I have addressed this below.

Validity of the findings

The data do appear to have been provided, however, I'd recommend consolidating the data into less files and providing metadata so others would clearly know what is what. Right now, the excel sheets are pretty messy and it's unclear what all the different numbers represent in some cases. I'd also recommend archiving the data (and ideally adding R code) in something like a Zenodo repository with a doi.

The conclusions are stated well and in line with the results. The discussion could benefit from some editing for flow, however. An introductory paragraph summarizing the results and general conclusions might be useful here.

Additional comments

Some of the supplementary information referenced in the text can't be accessed or found. Please double check this.

The authors switch between using "microparticles" and "pieces" throughout the text. I recommend sticking with one term. Particles or microparticles would be fine.

Abstract

Line 30: Anthropogenically modified natural materials are generally no considered to be microplastics, because they are not plastics. The term "anthropogenic particles" better captures all the particle types you list.

Lines 40-41: This line is confusing. What do you mean by "point sources" in this context? Suggest clarifying what this sentence means and why it is important.

Introduction

Lines 50-52: Microscopic debris is generated at all stages of product lifecycles depending on a multitude of factors that go beyond disposal. Suggest rewording.

Line 54: I'm unaware of microplastics being referred to as microparticles. If this is your term for this paper, then say so and provide a definition of microparticles. If this is a defined term in the literature, then explain and provide a citation. I don't think these terms can be used interchangeably, however.

Line 97: Technically, the difference would be "ontogenetically" not "temporally" for differences in life-stages at similar time points.

Methods

Line 109: I think a word is missing here. "These areas" perhaps?

Lines 112-114: Given that rockfish can display a high degree of site fidelity, it should be discussed at some point whether or not the fish would actually be expected to be making use of the reserves, as these distances are pretty large.

Line 152: It's worth noting that fume hoods are suboptimal for microplastics work because the suck air in rather than blow it out, so unclear why work was conducted in one. If it was just when working with hazardous chemicals make this clear.

Line 160: Why were all organs removed and measured if only GI tracts were analyzed?

Line 162: By "lining" do you mean the inside of the lids?

Lines 169-171: This is a unclear. So the liquid went through a 1 mm sieve, then a 63 micron, than onto a 5 micron PC filter? And the material on each sieve was analyzed separately? Where does the separatory funnel come in?

Line 183: You don't need the word "vacuum" here.

Lines 194-198: You haven't mentioned any blanks yet. This information should come after blanks are discussed. Also, it's not clear why this method would provide a good estimate for number of particles per fish. Study authors usually perform blank subtraction or set a limit of detection and/or quantification. See Brander et al. 2020.

Line 201: 30% per what? Were particles randomly selected? How?

Lines 204-206: Needs rewording.

Line 206: I believe it's a germanium diamond.

Lines 215-225: This whole paragraph needs careful editing and rewording. There should be enough information clearly presented so that someone else could exactly replicate the analyses if they had the same equipment and samples.

Line 219: What is math match? There will be some algorithm that compares a spectra with the library.

Lines 249-255: This information is important, but should be incorporated into the discussion as results are interpreted.

Lines 263-265: What do you mean by microplastic exposure? If you mean exposure over a full lifetime, then yes, this is an underestimate. However, with enough samples, even a snapshot should provide the range of microplastic particles present in a fish's digestive tract at any given time.

Line 268: This is not quite right. R is a programming language and is implemented in the R software environment. RStudio is an interface and isn't at all necessary for replications. So technically you would say you used "R" or the "R software environment" or something similar. Also, please indicate that the things in brackets are libraries.

Lined 272-274: Why Wilson's score intervals? This appears to be a method for binomial data, but you assumed Poisson distributions in your model. The model output in R should provide parameter confidence intervals so why not use those?

References: Rochman et al 2019 is duplicated

Results

Line 287: Thank you for providing these supplemental documents. However, Supplemental Document Figure 2 will not open. For simplicity, it would probably make more sense to put all the supplementary information into one word document.

Lines 296-298: Why was only a subset of particles measured? Which particles and how were they chosen?

Discussion

Lines 327-333: This information is already in the introduction and is not necessary to repeat here.

Line 383-384: This is pretty tenuous. A lot of those ropes would likely be plastic, such as polypropylene, and there are a lot of microfibres from textiles released by sewage outflow.

Lines 426-453: The concentrations of both microplastics and anthropogenic particles found by this study in the rockfish digestive tracts are likely quite low compared with the exposure concentrations in the toxicology studies discussed here. This should be discussed.

Figures

Figure 1: The 2nd to last sentence is confusing.

Figure 3: Are these values blank corrected?

Reviewer 2 ·

Basic reporting

The authors present a novel study on the occurrence of microplastics within the GI track of Black Rockfish collected from several sites along the Oregon coast. This is an important study that is well designed and written. However, it has several faults within the data presentation and the studies narrative. Specifically, the extent to which the authors do not account for the observed contamination of their samples and drastic overextensions as to what this data meets for human risk and marine reserves. Also, figure quality is exceptionally poor.
A major development in the field of microplastics has been our ability to detect and account for contamination, with several notable shifts in environmental concentrations occurring due to these developments. Currently, this study detects contamination but does not account for it beyond stating that it occurs and presenting a graph on the extent to which it may occur. Even the authors then discuss the occurrence of microplastics within GI tracks without consideration of the contamination. As sample processing occurred in batches, as did the processing of control samples, it seems like there is a much better route for considering the data that subtract controls from individuals, batches, sampling days, or regions.
Extending a novel and the importance of a study beyond the data could always be done with caution. The authors discuss at length the implications of this for human health, exposure, and policy. Several microplastics within the GI tract of Black Rockfish (which are rarely if ever eaten in North America) do not pose a risk to fish consumers. Furthermore, designing marine reserves is a complex socioecological topic that weights consider factors. Again, several microplastics within the GI tract of Black Rockfish should be a low priority within this space.
These statements are not meant to subtract from the importance of with work or the effort the authors have committed to completing this project. It is simply worth reflecting on issues this data allows us (the scientific community) to better understand. For me, the occurrence of microplastics within taxa is important. If the authors are keen to extend this data they could do so by noting that a key metric in toxicology is dose, which this data will inform. I am keen to see this manuscript published and have included several revisions below.


Abstract
1. Line 30:I will defer to the authors but I have not heard modified cotton referred to as a microplastic.
2. Line 31: “nearshore fish species in the Pacific Ocean” is a stretch given the study is on black rockfish from several sites along the Oregon coast.
3. Line 38: Following ‘12% of the particles were identified as synthetic using micro-FTIR.’ It could be worth stating the estimated number of microplastic per GI track (i.e. 12% x 93.1%). Via something like, Therefore, we estimate that …~11% of fish contain..
4. Line 47: I would remove ‘ and devise effective policy’. I appreciate the aim of having research influence policy but we (the scientific community) need to be honest about was does and doesn’t ‘move the needle’
5. Line 86: ‘to investigate life stage-specific microplastic consumption in’ requires revision if the aim is to set up your study as you do not look at consumption but focus on occurrence within the GI.
6. Line 96: Addition information on the aim of the study would be beneficial as many readers are selective in their focus.

Methods
7. Line 102: I applaud collaborating with anglers. Not a revision.
8. Line 113: Keep spacing before km consistent
9. Line 121: Additional detail of the difference should be included given the comparison of reserve/unprotected areas
10. Line 154: “We mostly followed…” requires further explanation
11. Line 184: Not needed in this case but Covernton et al. 2022 address this issue well using a Random Forest analysis, and does so in a similar system and taxa.

Covernton, G.A., Cox, K.D., Fleming, W.L., Buirs, B.M., Davies, H.L., Juanes, F., Dudas, S.E. and Dower, J.F., 2022. Large size (> 100‐μm) microplastics are not biomagnifying in coastal marine food webs of British Columbia, Canada. Ecological Applications, p.e2654.

12. Line 198: More detail required. Is this number then presented, subtracted from observed plastics, or just known? Ideally it should be adjusted for.
13. Line 246: Were particles matching the holders and nets excluded?
14. Line 250: I’m struggling to see how observed contamination was accounted for in this work, it seems that currently it was measured but not accounted for.

Results
15. Line 279: The 7.31 and 4.21 numbers should be included in the abstract.
16. Line 287: Again, stating this contamination is important but accounting for it is becoming common place within the field as accurate numbers are essential. Currently, the 7.31 and 4.21 numbers can not be assumed to be correct.

Discussion
17. Line 335: West Coast is incorrect given the area is the Northeast Pacific.
18. Line 345: The Covernton et al. 2022 mentioned above observed 1.12 (0.52–2.13) particles in black rockfish with full stomachs, compared to 0.10 (0.01–0.44) particles in the digestive tracts of rockfish with empty stomachs, which seems more relevant then the study discussed.
19. Line: 380: This is exactly the problem of not accounting for contamination. Here the author’s state ‘Microparticles across all types were consumed by Black Rockfish, at an average of 7.31 particles per subadult fish and an average of 4.03 particles per fish for the juveniles…’ this is not true. How can researchers, the public or managers be expected to account for contamination if the authors don’t even do it?
20. Line 437: I would suggest revising this statement considerably. This evidence does not exist currently, if it did, the some 200,000 microplastics people consume would warrant much great concern, but also, ~5.5 microplastics within the GI tract of a fish (which is rarely if ever consumed in North America) pose no risk to fish consumers. Really, this whole paragraph could be removed as the study does not contribute data to this topic.
21. Line 472: No reserve considers pollution is a bold state that I would suggest the authors consider.
Figures
22. Figures, I’d highly recommend exporting the images using code that allows the authors to specify the image size and resolution.
23. Figure 3: Given that sample processing occurred in batches, as did blank processing, it seems like there is a much better route for considering this data considered blank averages relative to individuals, batches, sampling days, or regions

Experimental design

Meets the standard but the analysis needs to account for contamination.

Validity of the findings

Fails to meet the standard until the contamination is addressed effectively.

Additional comments

I am keen to see this study undergo major revisions and be published.

Reviewer 3 ·

Basic reporting

The article «Presence of microplastics and microparticles in Oregon black rockfish sampled near marine reserve areas” by Lasdin, Arnold, Agrawal, Fennie, Grorud-Colvert, Atlesworth, Heppell and Brander. The authors have sampled subadult and juvenile Black Rockfish at the Oregon coast. Gastrointestinal tracts were analyzed to identify ingested microparticles.
Affiliations: United States/USA should be harmonized.
The abstract needs some improvement in phrasing. Did microplastic at the Oregon marine reserves also occur in juvenile fish?
I do not agree with the definition of "Microparticles in the Introduction, line 54: Microplastics are not more broadly referred as microparticles. Microparticles can also be of natural origin, organic or inorganic and can be of artificial origin but inorganic material. Please correct and employ a sharper phrasing.
Line 84: Here, studies on plastic that accumulate in underwater “barriers” such as basins and kelp forests could be cited.
Throughout the manuscript, there is a lack of discussing an important aspect of the occurance, uptace and toxicity of microplastics/particles: Their size. I will add several specific examples.
Line 169 and following: This paragraph needs better description. To me it is unclear: Were they sieved through 63 µm and then through 1 mm? That does not makes sense. And then: Was the filtrate or the content on the filter on a 5 µm filter? Why were the juvenile fish not filtered? What is the impact of the filtration/non-filtration on the detection limit?
Line 176 “Imaged all suspected plastics” – which guidelines were used? Later, Rochman is cited 182, rewrite more orderly.
Line 179…to ensure they were in the microparticle size range” – which is? Please elaborate.
Line 292: Results are described in a confusing way: 1000-1500 µm is most common and 500-1000 is also most common?
Line 301 “Enough qualifying spectra” – please report your criteria in a way the method can be reproduced. Furthermore, 131 passed the necessary threshold and 180 were run on FTIR is confusing. Are the sentences in the wrong order, or had 131+180 satisfying spectra? In line 308 “good matches” was added. Does that mean the same as “passed the threshold” or different? Please define these expressions beforehand and always use the same expression for describing the same phenomenon. That applies to more expressions, such as "microparticles" and "pieces" - it would be much easier to read if always the same word would be used to describe the same thing, especially when one expression is used for juvenile fish and another for subadult fish.
Line 312 is there more than one classification per one particle? If not – improve the semantics.
Please publish a supplement with the numbers of particles for each polymer type group identified, as information disclosure in addition to just “synthetic”.
Figure 7: Make clear if the hatched bars are stacked on top of the solid bars, or start at 0.
- Lines 279-280 – Viewed in context of contemporary plastics research and the title which reads “microplastics”, this way of presenting “microparticles” is misleading in my opinion. In the same sentence where numbers for the larger category “microparticles” are presented, also the number of clearly identified microplastics, which is only a fraction of the microparticles, should be presented. The same account for the statistical evaluation.
Line 284 – unclear between which 2 grouped locations.
Line 431 full stop missing
Line 459_ Where is Appendix T1?

Experimental design

The digestion time was rather long with rather high temperature. Potential impacts on the results (loss pf small particles of certain polymer types should be discussed, with references.
Line 196 Blank controls? How were they performed air/KOH/water sounds to me that no procedural blank was carried out where pre-filtered water is treated through all steps including initial air exposure, KOH treatment, filtering and analysis as if it were a sample. This needs to be done before the study is worthy of publication. Furthermore, more information on the other controls need to be provided, for example surface area and time exposed to the laboratory air.
- I do not understand the rationale of the authors dividing the number of sample microparticles by those of controls and multiplying by number of controls divided by the number of fish. The background data must be reported entirely with the entire sample results. The procedure needs to be described.
- In many figures, the average number of pieces per fish should be presented, with a confidence interval to get a representation of the robustness of the differences between the compared groups. Total number of pieces per location is confusing, as there were different numbers of fish per location investigated. Or did you digest all GITs of each location in one go? In that case, the numbers might be normalized to one fish?
- Also average of size per fish per location should be presented, and preferentially more physical data such as weight, maturation status, catch season and linked to background information such as season with typical spawning time etc., as some fishes do not eat during spawning. For juveniles mouth opening time compared to fish age and typical prey size is also important.

Figure 3: How was the average number of background calculated? The average background needs to be compared with the average fish content. Average number of pieces per fish with confidence intervals could say a bit more about the robustness of the data instead of showing the number for each fish. “pieces” and “per” to be separated. Harmonize nomenclature: Microparticles/pieces – same expression for same phenomenon. To describe the results gained by this method as “consumed by each fish” is interpreting too much into highly uncertain data. “Microparticles detected in each GIT” would be more correct.
Figure 4: Size needs to start at detection limit, not at 0. Name smallest particle you found. Average number of pieces per fish with confidence intervals could say a bit more about the robustness of the data instead of total number of pieces in this study
Figure 5: Average number of pieces per fish better.
Supplemental figure 1: Average number of pieces per fish with confidence intervals could say a bit more about the robustness of the data instead of total number of pieces in this study.

Even better: Can you do a recovery experiment to determine the smallest size you can semi-quantify?

Validity of the findings

Newport subadults were dissected in the field, whole reserve fish samples remained frozen. Wiped with EtOh prior to dissection - potential contamination of the Newport subadults by air needs to be discussed in relation to the data.
Figure 2: Please present the normal distribution of dataset. If not normally distributed, Anova is unsuited and a non-parametric test should be applied. What do the whiskers represent?
- Important weaknesses were not discussed. I suspect, for example, that the two lowest size classes (0-500 µm) were not actually present in lower numbers in the samples, but only detected in lower numbers due to poor method quality missing out on the smaller particles. It is published that these particles are difficult to find manually (for example Lusher et al.), and that should be discussed. As in line 366-368 the size distribution is inticated (should be result), the lower detection limit/recovery should be determined and discussed, as the minimum numbers might rather represent methodological limitations than that situation in the environment.
396-403 It needs to be clearly stated that comparisons can only work within size classes and with defined measurement uncertainties and limits of detection.
There are several reporting guidelines mentioned in Cowger 2020, where the last author of this manuscript is a co-author, that were not heeded in this manuscript, such as error propagation, limit of detection etc. A thorough following of their own guidelines would lift the quality of this manuscript.

Line 437: “There is evidence that microparticles could have deleterious effects on fish consumers: Are the GITs that were investigated consumed by humans? If not, this sentence is not related to this study. This sentence stand also alone without any reference and is followed by fish health study discussions rather than human health. The paragraph ends with descriptions in human food, but the present study does not investigate parts of fish that are consumed? Also are juveniles and sub-adults consumed at all? Here, ecosystem or population stability discussions may fit better than food safety evaluations?

Conclusions are vague: “not..a consensus as to what values are biologically relevant” - what is meant by values? *Numbers, type, size? Biologically relevant (for which endpoint)? There is a growing body of evidence of effects measured in exposure studies. Those have not been discussed. Also, this is not really a conclusion of the findings. Much more interesting is the finding of higher pollution in the protected area. (if the statistics hold)

Conclusions do not add significant new knowledge. “Microplastics were detected…in the gastrointestinal tracts…demonstrated that plastic is indeed found in the nearshore waters… No attempt of verifying the methods or measurement uncertainty or recovery has been attempted. If that could be added, the study would gain impact.

Additional comments

Discussion is a bit lengthy, reducing to only those points that can be related to the data presented – i.e. similar type, shape, size range, and concentration when presenting toxicity studies would be useful. The field of microplastics research is repeating a lot of data in introductions and discussions which is tedious to read could be improved if one first has the idea. I hope this can be done.

---

## Round 0.2 · Minor Revisions

· Academic Editor

Minor Revisions

The reviewers appreciated your efforts in revising the document. There are a couple more comments that they would like you to address. For example, there is a possibility that the numbers of microplastics detected may be massively underestimated as a rather high concentration of KOH in combination with a rather high temperature and exposure time has been applied to degrade the fish samples. Also, some further clarification on the description of microplastics and microparticles.

Reviewer 1 ·

Basic reporting

No comment.

Experimental design

No comment.

Validity of the findings

No comment.

Additional comments

Thank you for addressing all the review comments. The manuscript is improved. I have a few further suggestions below, but the manuscript is otherwise close to being suitable for publication. One crucial point, is for the authors to make their definitions of what microplastics vs. microparticles vs. anthropogenic particles are extremely clear and to be incredibly careful about using the right word throughout the article as they are discussing their findings. For example, microplastic would only be appropriate when discussing chemically confirmed (or corrected for % plastics) particles.

Lines 28-30: This description of microplastics and microparticles is still not satisfactory. The first sentence in the abstract discusses microplastics and then then the next sentence states that "These microparticles are known to encompass synthetic, semi-synthetic and anthropogenic particles." The definitions of each of these are unclear and conflicting. The definition of microplastics in fact does not include semi-synthetic and anthropogenic particles. But anthropogenic particles can include microplastics, and it's not clear whether microparticles and anthropogenic particles are the same thing or not. Are microparticles any small particle, including sand? It also then doesn't make sense to say "microplastics and microparticle" because microplastics are already included in the category of microparticle. I recommend starting broad by introducing anthropogenic particles, including microplastics.

Lines 33-34: References to figures or supplemental materials are not provided in an abstract. The abstract should read as a succinct description of the study that does not require the reader to refer to anything else.

Lines 364-367: This sentence describes similar average numbers of microplastics consumed by adult and juvenile fish but then provides percentages instead of particle counts. Are these meant to be rates of microplastic occurrence? If so, the wording needs to be changed.

Lines 438-439: This sentence is confusing.

Figure 2: Please indicate in the caption what the blue dashed line represents.

Reviewer 3 ·

Basic reporting

The phrasing, sorting, and statistical questions have been addressed sufficiently according to my requests. Thank you for taking the time to make the article more readable and more precise. There is still room for improvement, for example, size of letters in figures.

Experimental design

I am hesitantly ok with the background reporting now.

Validity of the findings

However, my main concern was not sufficiently addressed, or maybe even in a misleading way, which I hope was not on purpose. There is a possibility that the numbers of microplastics detected may be massively underestimated as a rather high concentration of KOH in combination with a rather high temperature and exposure time has been applied to degrade the fish samples. The two references the authors listed in the author’s response (“based off other studies including several studies that have used higher temperatures including Caldwell et al 2022 (65˚C) and Li B 2020 (65˚C)”) that the authors write they used to base their method on, did not do any recovery testing so are not valid for this argument. One of them is an exposure study only using polystyrene. Also, the several times over copied paragraph about their participating in a ring test in detecting microplastics in water does - while recommendable for their end-point analysis – not say anything about the potential loss of microplastic particles in the degradation step. Unless several microplastic types (as the different polymers have different susceptibility to disintegration) of several sizes 20 µm to 500 µm (as smaller particles degrade faster due to a high surface to volume ratio and then are lost through the pores in filtration) have been tested for their withstanding this treatment, I continue to suspect that the two lowest size classes (0-500 µm; Figure 4) were not actually present in lower numbers in the samples, but only detected in lower numbers due to poor method quality missing out on the smaller particles.
Regarding “We did do a recovery experiment separately, and our size detection limit is 20 um. This is described in Defront et al. (2022).” I could not find this reference and it is not cited in the manuscript. I guess it is “De Frond et al. 2022”… In this article, microplastic in clean water is investigated, and no KOH degradation is involved. There are very much fewer challenges in water than in complex matrices, this cannot be used as a reference for recovery in a complex matrix.

Additional comments

Unless this (lack of valid recovery study, uncertainty of numbers with potential loss through KOH degradation, especially for small particles and different susceptibility of different polymer types) is really clearly discussed, or preferably a valid recovery study carried out, the article should not be published.

---

## Round 0.3 · accepted · Accept

· Academic Editor

Accept

Thank you for addressing this last round of comments. I appreciate you addressing the issue of recovery by the inclusion of additional text and a study limitations section.